# Integrating phytochemical analysis and experimental validation of *Ziziphus jujuba* seed powder and oil to ameliorate CCl$_4$-induced liver fibrosis in sprague dawley rats

Laiba Tanvir[1], Khurram Afzal[1]*, Asad Abbas[1]*, Adnan Amjad[1], Talha Bin Iqbal[2], Naeem Sarwar[1], Rameen Naeem[1], Ayman Furqan[1], Bipindra Pandey[3]*

1 Department of Human Nutrition, Faculty of Food Science and Nutrition, Bahauddin Zakariya University, Multan, Pakistan, 2 Department of Food Science and Technology, Faculty of Food Science and Nutrition, Bahauddin Zakariya University, Multan, Pakistan, 3 Department of Pharmacy, Madan Bhandari Academy of Health Sciences, Hetauda, Bagmati Province, Nepal

* bipindra.pandey@mbahs.edu.np (BP); Khurram.afzal@bzu.edu.pk (KA); asadabbaskhichi@gmail.com (AA)

## Abstract

Liver fibrosis, a consequence of chronic liver injury, is characterized by excessive accumulation of the extracellular matrix, oxidative stress, and activation of hepatic stellate cells. This study evaluated the effectiveness of *Ziziphus jujuba* (*Z. jujuba*) seed powder and seed oil, both separately and in combination, in alleviating CCl$_4$-induced liver fibrosis, and linked the in vivo effects to their phytochemical profiles. The seed powder extracts, using acetone and ethanol, showed significant antioxidant activity, with DPPH values reaching up to 90.13%, and exhibited a high phenolic content of 79.57 mg GAE/g in the ethanol extract. Gas chromatography–mass spectrometry (GC–MS) analysis identified several bioactive compounds, including derivatives of 1,4-benzenedicarboxylic acid (30.46% in powder, 20.43% in oil), hexanedioic acid bis(2-ethylhexyl) ester (25.31% in powder, 20.78% in oil), and oleic acid (10.61% in powder, 16.06% in oil). In vivo, carbon tetrachloride (CCl$_4$) administration led to elevated levels of ALT, AST, ALP, and bilirubin, causing disruptions in oxidative, lipid, inflammatory, hematological, and nutritional parameters. All treatment groups showed improvements in these parameters, with Group 4 (G$_4$) exhibiting the most pronounced hepatoprotective effects, including reductions in ALT (75.56 U/L), AST (125.76 U/L), ALP (125.43 U/L), and bilirubin (0.40 mg/dL) levels. Oxidative stress markers were reduced, with MDA at 5.47 nmol/g protein and NOx at 46.81 nmol/g protein, whereas antioxidant defenses were enhanced, as evidenced by SOD activity at 71.69 U/mg and catalase restoration. Lipid profiles were normalized, with triglycerides (TG) at 80.01 mg/dL, HDL at 34.60 mg/dL, and LDL at 22.30 mg/dL. Additionally, cytokine levels, specifically TNF-α and IL-6, decreased, red and white blood cell differentials were restored, and feed and water intakes and serum protein levels improved. These

---

**Data availability statement:** All relevant data are within the paper.

**Funding:** The author(s) received no specific funding for this work.

**Competing interests:** The authors declare that no competing interests exist.

**Abbreviations:** *Z. jujuba*: *Ziziphus jujuba*; CCl$_4$: Carbon Tetrachloride; GC-MS: Gas Chromatography-Mass Spectrometry; ALT: Alanine Aminotransferase; AST: Aspartate Aminotransferase; ALP: Alkaline Phosphatase; MDA: Malondialdehyde; NOx: Nitric Oxide Metabolites; CAT: Catalase; HDL: High-Density Lipoprotein; LDL: Low-Density Lipoprotein; VLDL: Very Low-Density Lipoprotein; WBC: White Blood Cell; RBC: Red Blood Cell; HCT: Hematocrit; TBARS: Thiobarbituric Acid Reactive Substances; TGF-β: Transforming Growth Factor Beta; NF-κB: Nuclear Factor Kappa B.

findings highlight the synergistic anti-fibrotic and hepatoprotective properties of *Z. jujuba* seed powder and oil, likely facilitated by the bioactive compounds in both the polar and lipid phases. However, further research into histopathology and confirmation of molecular pathways is crucial for clinical application.

## Introduction

Liver fibrosis is a complex pathological condition that arises from persistent and chronic liver injury. It is characterized by an excessive buildup of extracellular matrix (ECM) proteins, such as collagen, which disrupts the normal architecture of the liver and impairs its functionality [1,2]. This condition is widely considered an intermediate stage in the progression toward cirrhosis, liver failure, and hepatocellular carcinoma, all of which are associated with significant morbidity and mortality globally [3]. Unlike acute liver injury, fibrosis is a dynamic process that involves a continual wound healing response that fails to resolve, resulting in pathological scarring [4]. The growing incidence of liver fibrosis is strongly associated with the increasing prevalence of metabolic diseases, viral infections, and alcohol-related liver disease, particularly in low- and middle-income countries. This trend highlights the urgent need for effective therapeutic interventions to address this escalating health challenge [5].

Hepatic stellate cell (HSCs) activation is a key pathological feature of liver fibrosis. Under normal conditions, these cells remain quiescent and primarily function as vitamin A storage cells. However, upon liver injury, HSCs transform into myofibroblast-like cells that produce fibrillar collagens and other ECM components, thereby exacerbating tissue scarring and disrupting the hepatic microarchitecture [6,7]. The molecular and cellular mechanisms underlying HSC activation involve inflammatory signaling pathways, including cytokines such as tumor necrosis factor-alpha (TNF-α) and interleukin-6 (IL-6), as well as oxidative stress mediated by reactive oxygen species (ROS). Additionally, immune cell recruitment contributes to an inflammatory environment that perpetuates fibrosis [8,9]. The progression of liver fibrosis also involves dysregulated signaling pathways, such as TGF-β, platelet-derived growth factor (PDGF), and the gut–liver axis, which promote fibrogenic responses and ECM remodeling [10–12]. Despite significant progress in understanding these underlying mechanisms, effective anti-fibrotic therapies remain elusive. Current therapeutic strategies primarily focus on addressing the root cause of liver fibrosis, rather than directly reversing the condition.

Historically, medicinal plants have played a pivotal role in managing liver diseases owing to their bioactive compounds, which possess antioxidant, anti-inflammatory, and immunomodulatory properties [13–15]. Among these plants, *Z. jujuba* (common name: Unaab) has attracted significant attention because of its rich phytochemical composition and wide range of pharmacological activities. This deciduous tree, native to Asia and cultivated across subtropical regions, produces seeds that are abundant in flavonoids, saponins, triterpenic acids, and unsaturated fatty acids, including oleic and linoleic acids. These compounds contribute to the antioxidant and

hepatoprotective effects of the plant [16,17]. Research has shown that compounds extracted from *Z. jujuba* seeds, such as jujubosides, modulate key inflammatory and fibrogenic pathways, including TLR4-MyD88 and NF-κB signaling. This modulation leads to a reduction in hepatic damage and fibrosis in experimental models, supporting the plant's potential as a therapeutic agent for liver disease [18].

Despite extensive research on the antioxidant and anti-inflammatory benefits of *Z. jujuba* seed derivatives, there are limited comparative studies on the therapeutic potential of its seed oil versus seed powder in the context of liver fibrosis. Seed oil, which is rich in lipid-soluble antioxidants and essential fatty acids, may exert cellular effects compared to the complex phytochemical matrix found in seed powder [19,20]. Given the global increase in liver fibrosis, there is an urgent need for novel therapeutic options, such as *Z. jujuba* seed powder and oil, which have shown promise in preclinical studies. This rise in liver disease underscores the importance of natural food products, especially those enriched with phytochemicals, in reducing the global burden of liver disease. Understanding the differential effects of *Z. jujuba* seed oil and powder is crucial for optimizing their use as nutraceuticals or adjunctive therapies for chronic liver diseases. The present study aimed to evaluate the anti-fibrotic potential of *Z. jujuba* seed powder and oil in a rat model of $CCl_4$-induced liver fibrosis. Phytochemical analysis was conducted using GC–MS, and the therapeutic efficacy was assessed using biochemical, oxidative stress, and histopathological parameters (Fig 1).

## Materials and methods

### Raw material procurement

Seeds of *Z. jujuba* Mill. were procured locally from Multan, Pakistan. Their authenticity was confirmed by experts from the Department of Food Science and Nutrition, Bahauddin Zakariya University, Multan. The seeds were washed several times with distilled water to remove any impurities. The samples were then dried in a hot air oven (Model BJ-9176) at 40°C for 48 h until the moisture was removed. Dried seeds were ground to a coarse powder using a mechanical grinder, sieved to obtain a uniform particle size, and stored in airtight containers at 4°C until use. Seed oil was extracted via cold pressing using a mechanical oil press (Model GD-HOPM-150) at ambient temperature to preserve the bioactive compounds. The extracted oil was filtered and stored in amber bottles at 4°C.

### Phytochemicals analysis

To evaluate the antioxidative potential of *Z. jujuba* seed powder and oil, a series of in vitro assays was conducted. For the extraction process, 10 g of seed powder was added to 100 mL of solvent in a 1:10 (w/v) ratio. The solvents used were ethanol, hexane, acetone, and distilled water. Each mixture was placed in a conical flask and agitated continuously on an orbital shaker for 36 h to ensure the maximum extraction of antioxidant constituents. For the oil samples, 1 mL of oil and 5 mL of either ethanol or methanol were mixed and vortexed thoroughly to ensure complete homogenization. After extraction, all samples were filtered using filter paper (Whatman No. 4), and the resulting filtrates were concentrated using a rotary evaporator (Model: RE202).

### DPPH for free radical scavenging activity

The antioxidant properties of *Z. jujuba* seed powder and oil were evaluated using the DPPH radical-scavenging technique [21]. A 0.5 mL sample of *Z. jujuba* seed powder extract and *Z. jujuba* seed oil was mixed with 3 mL of DPPH solution and then diluted with 2 mL of methanol. These mixtures were kept in the dark for a set period, after which their absorbance was recorded at 517 nm using a UV–visible spectrophotometer (Model no: 823−0210 P-2-R) to assess their free radical scavenging ability.

$$DPPH\ \% = \frac{Sample\ absorbance - Blank\ absorbance}{Control\ absorbance} * 100$$

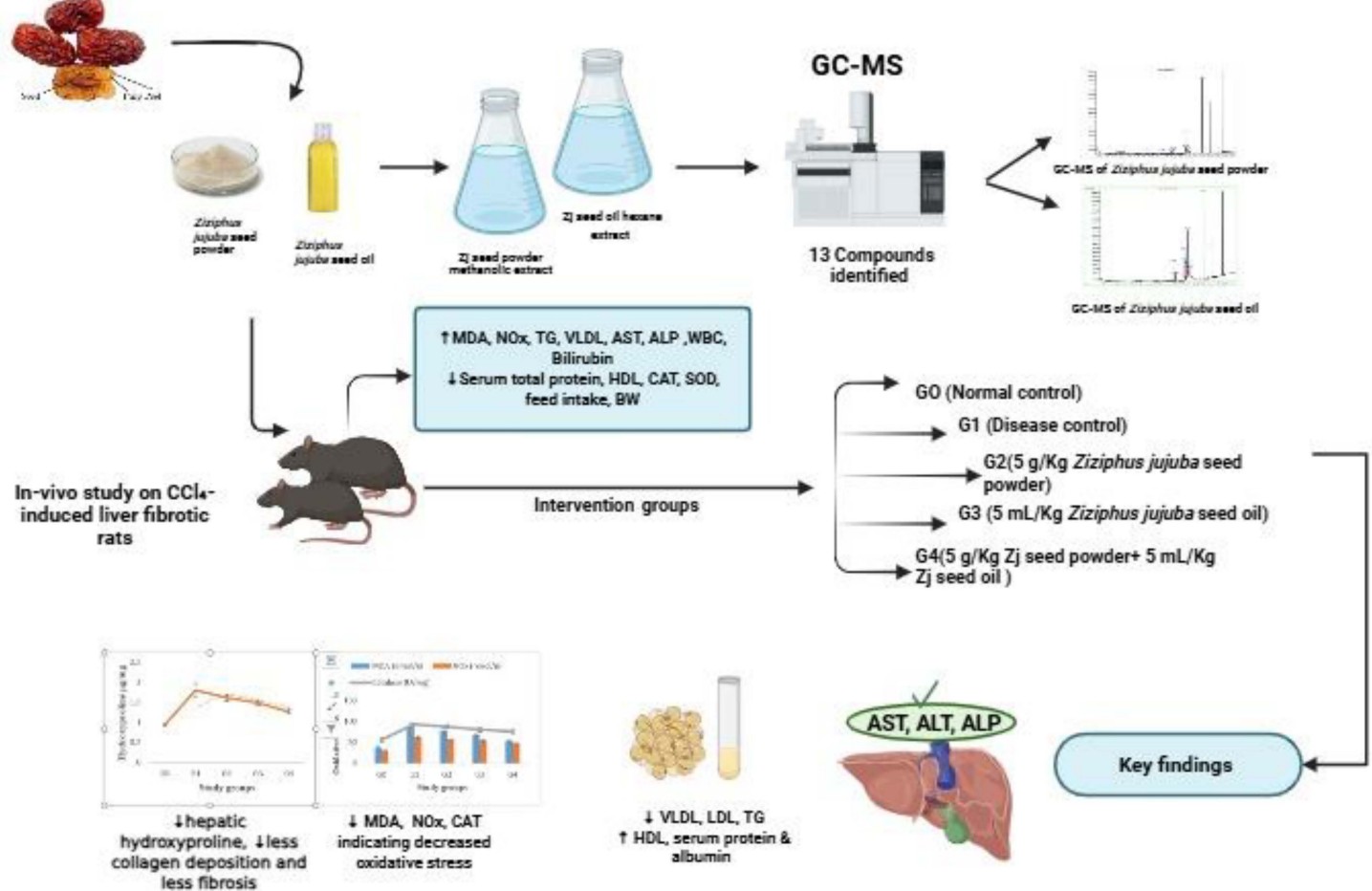

**Fig 1. Schematic representation of the effects of *Ziziphus jujuba* seed powder and seed oil on CCl4-induced liver fibrosis.**

## Ferric reducing antioxidant power (FRAP)

The antioxidant capacity of *Z. jujuba* seeds and their oils was evaluated using the ferric reducing antioxidant power (FRAP) assay [22]. To conduct the test, FRAP reagents, including $FeCl_3 \cdot 6H_2O$, acetate buffer, and TPTZ solution, were combined with the *Z. jujuba* seed powder and oil. The mixture was then incubated at 37 °C in the dark for 30 min. The absorbance of each sample was recorded at 593 nm using a spectrophotometer (Model 823−0210 P-2-R).

## Total phenolic content (TPC)

The total phenolic content of *Z. jujuba* seed powder and seed oil extracts was determined using the Folin–Ciocalteu method, and the results were expressed in mg GAE/g [23]. All mixtures were kept in a dark room for 1 h, and the absorbance of both the seed powder and seed oil extracts was determined using a spectrophotometer at 760 nm (Model no: 823−0210 P-2-R).

## Quantification of phytochemicals of seed powder and seed oil of *Z. jujuba* by GC/MS

The phytochemical components of the methanolic extract of *Z. jujuba* seeds and the hexane extract of its oil were analyzed using gas chromatography-mass spectrometry (GC-MS) (model number: Agilent 5977 B GC/MSD) with a thermal

desorption system [24]. The gas chromatography column used had dimensions of 30 m length, 0.25 mm internal diameter, and 0.25 μm film thickness. The initial temperature of the column was set at 100 °C and increased by 5 °C per minute until it reached 280 °C, where it was maintained for 3 min. Subsequently, the temperature was increased to 280 °C at a rate of 15 °C/min, with a 35-minute isothermal hold. Helium was used as the inert carrier gas at a flow rate of 1.41 mL/min. Samples diluted to 1% v/v were introduced with an injection volume of 1 μL in splitless mode. The mass spectrometer was operated at an ion source temperature of 230 °C and an interface temperature of 270 °C. Data collection was conducted in full-scan mode over a mass-to-charge (m/z) range of 40–650. The volatile compounds from the *Z. jujuba* seed powder and seed oil were identified by matching mass fragmentation data (MS), Kovats indices (KI), and chromatographic peaks with reference spectra from the NIST 20 (National Institute of Standards and Technology) Mass Spectral Library, RTLPEST 3 libraries, and the 4th edition of Adams [59] and by comparison with the previously published literature [60,61].

## Study animal, experimental design, and dose selection

Forty male Sprague-Dawley rats, weighing–180–220 g and aged 8–10 weeks were obtained from the animal facility at the Department of Pharmaceutical Sciences, BZU, Multan. The rats were kept in polypropylene cages, with eight rats per cage, under controlled environmental conditions: a temperature of $22 \pm 2$°C, relative humidity of 50–60%, and a 12-hour light/dark cycle. Throughout the study, the animals had unrestricted access to standard basal diet and water. The experimental design included five groups and eight male Sprague-Dawley rats in each group: normal control ($G_0$) and disease control ($G_1$), while the intervention groups received a standard basal diet supplemented with *Z. jujuba* seed powder at 5 g/kg ($G_2$), seed oil at 5 mL/kg ($G_3$), or their combination at 5 g/kg + 5 mL/kg ($G_4$). *Z. jujuba* seed powder and seed oil was administered daily via oral gavage tube mixed with standard basal diet during the experimental period in the experimental animals. *Z. jujuba* seed powder and seed oil were dissolved in 0.5% (w/v) sodium carboxymethyl cellulose, which was used as a vehicle during the administration to–$G_2$-$G_4$ and orally administered at a dose of 10 mL/kg for the study period. However, for the $G_0$ and $G_1$ groups, only 0.5% (w/v) sodium carboxymethyl cellulose was administered orally at a dose of 10 mL/kg body weight of experimental animal. The dosages of *Z. jujuba* seed powder and seed oil were established based on the existing literature on the clinical practices of Chinese medicine practitioners [62]. The human-equivalent lower dose is 0.81 g/kg, which corresponds to 3.2 times the clinical dosage, whereas the human-equivalent higher dose is 3.23 g/kg, equating to 12.9 times the clinical dosage [63,64] (Table 1).

## Induction of liver fibrosis

Liver fibrosis was induced by administering carbon tetrachloride ($CCl_4$) via intraperitoneal injection at a dosage of 1 mL/kg, mixed in a 1:1 ratio with olive oil. This was performed twice a week for 6 weeks, totaling 12 injections [25]. The treatments were initiated simultaneously with $CCl_4$ injections.

**Table 1. Experimental design of the in vivo study.**

| Treatment Groups | Interventions and dosage administration |
| --- | --- |
| $G_0$ | NC group |
| $G_1$ | DC group |
| $G_2$ | SBD + *Z. jujuba* seed powder(5 g/kg) |
| $G_3$ | SBD + *Z. jujuba* seed oil (5 mL/kg) |
| $G_4$ | SBD + *Z. jujuba* seed powder + *Z. jujuba* seed oil (5 g/kg + 5 mL/kg) |

$G_0$, Normal control (no disease induced); which received only the standard basal diet. $G_1$, Disease control group ($CCl_4$-induced), which was diseased and received the standard basal diet; $G_2$ represents the group treated with *Z. jujuba* seed powder (5 g/kg); $G_3$ received Z. jujuba seed oil (5 mL/kg); $G_4$ was given a combination of *Z. jujuba* seed powder (5 g/kg) and seed oil (5 mL/kg) along with the standard basal diet.

## Ethical statements, sample collection, and Humane endpoint

The Institutional Review Committee approved the experimental protocol (Approval No: BZU/IAEC/2024/032) used in this study involving experimental animals. Before the experiments began, all animals were randomly assigned to different groups. This research strictly followed the National Institutes of Health (NIH) regulations and guidelines [58]. Efforts were made to minimize animal distress in line with the principles of the National Center for the Replacement, Refinement, and Reduction of Animal Research (NC3Rs), as detailed in the Animal Research Reporting of In vivo Experiments (ARRIVE) guidelines (S1 File). Following the treatment phase, the rats were deprived of food overnight and anesthetized with a ketamine/xylazine mixture (80/10 mg/kg). Blood was drawn via cardiac puncture using sterile syringes, left to clot, and centrifuged at 3000 rpm for 10 min to obtain the serum. The serum samples were stored at −20°C for later biochemical analysis [26]. Animals were euthanized by cervical dislocation, and their livers were quickly removed, washed with ice-cold saline, weighed, and divided for biochemical testing.

## Body weight measurement and food consumption

The body weights of the rats were recorded weekly and at the end of the study. Relative liver weight was calculated as the percentage of liver weight to final body weight. The food intake of the animals in the different groups was observed weekly. The per capita water and food consumption for each week was calculated using the following formula:

$$\text{Per week water and feed consumption} = \frac{\text{Food given (g)} - \text{Leftover food (g)}}{\text{Animals numbers per group}}$$

## Oxidative stress markers in liver tissue

Liver tissue samples (0.25 g) from each experimental group were homogenized in 1 mL of ice-cold phosphate-buffered saline (PBS) with 0.1 M EDTA at pH 7.4. The homogenized mixtures were centrifuged at 12,000 × g for 20 min at 4 °C. The obtained supernatant was used for biochemical analysis. The protein concentration in the tissue homogenates was determined using the Bradford assay with bovine serum albumin as the standard [27].

## Malondialdehyde (MDA) determination

The levels of MDA, which serves as a marker for lipid peroxidation, were assessed in 10% liver homogenates prepared using a 0.9% NaCl solution through a thiobarbituric acid reactive substances (TBARS) assay [28].

## Catalase activity

Catalase activity was evaluated by observing the rate of hydrogen peroxide ($H_2O_2$) breakdown. A small amount of liver supernatant (1 μL) was mixed with 1 mL of 2 mM $H_2O_2$ in a 0.1 M PBS solution at pH 7.0. The reduction in absorbance at 240 nm was measured every 30 s for 2-minute period. Catalase activity was determined using a molar extinction coefficient of 43.6 $M^{-1}cm^{-1}$ and reported as units (U) per milligram of protein [29].

## Nitric Oxide (NOx) level assay

To indirectly estimate NO levels, total nitrate and nitrite concentrations, which are stable metabolites of NO, were measured. The assay mixture comprised 100 μL of liver homogenate, 400 μL of Griess reagent (1% sulfanilamide and 0.1% N-(1-naphthyl) ethylenediamine dihydrochloride in 2.5% phosphoric acid), and 500 μL of distilled water. The samples were incubated at room temperature for 10 min to facilitate color development. The absorbance was immediately measured at 540 nm using a spectrophotometer. The concentration of nitric oxide was determined from a sodium nitrite standard curve and expressed in μmol per mg of protein [30].

### Superoxide Dismutase (SOD)

Superoxide dismutase activity was evaluated by measuring its ability to decrease nitrotetrazolium chloride photochemical reduction under a controlled environment. The superoxide anions produced by riboflavin-mediated illumination react with SOD and decrease formazan product formation. The assay utilized 10 µL of liver tissue homogenate mixed with 641 µL of 0.067 M phosphate-buffered saline (pH 7.0), 40 µL of 0.1 M EDTA, 20 µL of 1.5 mM nitro blue tetrazolium (NBT), and 9 µL of 0.1 mM riboflavin. The mixture was stirred gently and illuminated with a 40-W lamp at a distance of 15 cm for 15 min. The absorbance was measured immediately at 560 nm using a spectrophotometer. One SOD unit represented the enzyme quantity required to inhibit 50% of NBT reduction within 1 min at 25°C, expressed as U/mg protein [68].

$$\%\text{Inhibition} = \frac{\text{Absorbance of control} - \text{Sample absorbance}}{\text{Control}} \times 100$$

### Liver function tests and inflammatory markers

Blood samples were obtained in vials coated with EDTA for hematological testing and gel vials without anticoagulants for serum biochemical analysis [31]. To separate the serum, clotted blood was centrifuged at 3500 rpm for 10 min using a centrifuge (Model # Z326K), and the serum was then stored at −20 °C until needed. Hematological parameters, such as total white blood cell (WBC) count and its differentials, were assessed using an automated hematology analyzer (Cobas 6000). Serum biochemical markers, including alanine aminotransferase (ALT), aspartate aminotransferase (AST), alkaline phosphatase (ALP), bilirubin, urea, uric acid, and creatinine, were evaluated using automated clinical chemistry analyzers (Lisa 300 Hycel automation). Moreover, inflammatory markers (TNF-α and IL-6) were measured using an enzyme-linked immunosorbent assay (ELISA) kit using a microtiter plate reader. The absorbance was measured at 450 nm, and the concentration of these anti-fibrotic cytokines was expressed as pg/mL [71].

### Lipid and lipid profile

The lipid profile, which encompasses high-density lipoprotein (HDL), total cholesterol (TC), low-density lipoprotein (LDL), very low-density lipoprotein (VLDL), and triglycerides (TG), was assessed using standard enzymatic techniques and calculation methods widely recognized in clinical settings. Total protein concentration was determined using a colorimetric assay. These measurements provide vital insights into cardiovascular risks and metabolic health [32].

### Statistical analysis

Each test was conducted three times, and the results are presented as the mean ± standard deviation (SD). To compare the groups statistically, a one-way analysis of variance (ANOVA) was utilized, followed by the Least Significant Difference (LSD) post-hoc test, using SPSS software version 16. A p-value of less than 0.05 was considered statistically significant.

## Results

Key findings of the study investigating the therapeutic effects of *Ziziphus jujuba* seed powder and seed oil on CCl$_4$-induced liver fibrosis in rats. This includes the analysis of multiple biochemical, hematological, and physiological parameters, emphasizing antioxidant activity, liver function, hematology, and metabolic profiles, with data derived directly from group-wise experimental measurements. Additionally, water and feed intakes were observed throughout the 4-week trial.

### Antioxidant activity of *Z. jujuba* seed powder and oil

The antioxidant capacity of *Z. jujuba* seeds was confirmed using multiple assays, including DPPH radical scavenging, TPC, and FRAP (Table 2). Seed extracts, particularly those prepared with acetone and ethanol, exhibited higher radical

**Table 2. Antioxidant activity of *Z. jujuba* seed powder and oil.**

| Plant | Solvent | DPPH (%) | TPC (mg GAE/g) | FRAP (µmol Fe²⁺/g) |
|-------|---------|----------|----------------|---------------------|
| **Oil** | Acetone | 75.28±0.54[c] | 67.05±3.65[d] | 667.7±8.53[d] |
| | Ethanol | 68.17±0.05[e] | 72.68±2.38[c] | 870.7±5.24[b] |
| | Hexane | 59.31±0.38[f] | 48.45±4.28[h] | 503.0±6.49[f] |
| | Water | 56.30±1.42[g] | 60.66±4.37[f] | 736.7±4.58[c] |
| **Seed** | Acetone | 90.13±6.78[a] | 74.21±5.63[b] | 1041.2±9.31[a] |
| | Ethanol | 81.16±7.53[b] | 79.57±4.51[a] | 601.5±4.68[e] |
| | Hexane | 49.62±6.03[h] | 57.21±3.62[g] | 479.3±8.23[f] |
| | Water | 72.60±4.63[d] | 63.58±3.64[e] | 589.0±5.53[e] |

*Means having same alphabets in columns do not differ significantly (P-value < 0.05).*

scavenging activity (up to 90.13%) than seed oil extracts, demonstrating the abundance of polar antioxidant phytochemicals, such as flavonoids and phenolics, in the seed matrix. Hexane extracts exhibited lower antioxidant activity, reflecting the limited capacity of the solvent to solubilize polar compounds.

TPC analysis indicated that the ethanol-extracted seed powder possessed the highest phenolic content (79.57 mg GAE/g), which correlated with its strong antioxidant activity. Seed oil phenolics were present at lower levels but contributed significantly to the overall antioxidant potential. FRAP assays supported these findings, with the acetone seed extract showing a superior reducing power.

### Phytochemical compounds identification of *Z. jujuba* seed powder by gas chromatography-mass spectrometry (GC-MS)

Gas chromatography-mass spectrometry (GC-MS) analysis of the methanolic extract of *Ziziphus jujuba* seed powder identified several bioactive compounds with varying abundances. The extract was dominated by 1, 4-Benzenedicarboxylic acid derivatives (30.46%) and Hexanedioic acid, bis(2-ethylhex) (25.31%), followed by Bis(2-ethylhexyl) phthalate and related phthalates (17.59%), and Oleic acid (10.61%). These major constituents suggest the presence of fatty acids, esters, and phthalate derivatives, which are often associated with antioxidant, antimicrobial, and plasticizer properties. Minor compounds, such as n-hexadecanoic acid (2.63%), octadecanoic acid (2.38%), and various unsaturated fatty acids (linoleic and linolenic acid derivatives, 3.62%), contribute to the nutritional and pharmacological potentials of the extract. The relatively high proportion of fatty acid esters and phthalate derivatives indicates that the seed extract is rich in lipid-soluble bioactive molecules, which may play a role in its traditional medicinal use (Fig 2; Table 3).

### Phytochemical compounds identification of *Z. jujuba* seed oil by gas chromatography-mass spectrometry (GC-MS)

GC–MS analysis of the hexane extract of *Z. jujuba* seed oil revealed the presence of diverse fatty acids, esters and phthalate derivatives. The extract was primarily dominated by hexanedioic acid and bis(2-ethylhexyl) (20.78%) and 1,4-Benzenedicarboxylic acid derivatives (20.43%), which together accounted for over 40% of the total composition. Another major component was Oleic acid (16.06%), a monounsaturated fatty acid known for its nutritional and therapeutic value. Moderate proportions of phthalate derivatives (7.81%), methyl esters of linoleic and linolenic acids (4.59–4.85%), and n-hexadecanoic acid (4.23%) were also identified. Several minor constituents, including methyl stearate derivatives (1.34%) and octadecatrienoic acid methyl esters (<1%), contributed to the lipid profile (Table 4). *Z. jujuba* seed oil is rich in unsaturated fatty acids (oleic, linoleic, and linolenic acids), which are nutritionally beneficial, while the presence of phthalate-related compounds suggests possible environmental or solvent-related contamination (Fig 3; Table 4).

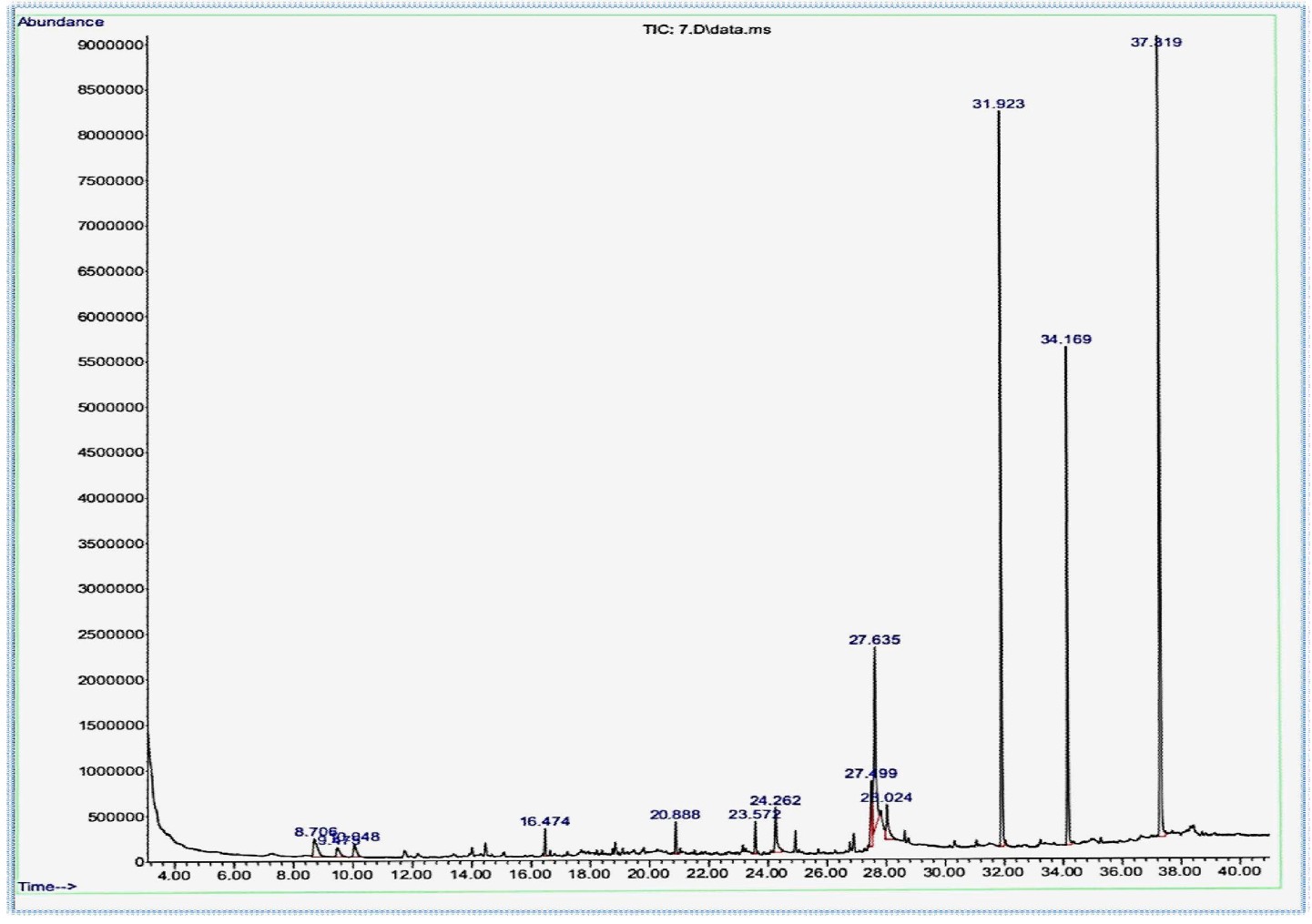

**Fig 2. GC-MS chromatogram of the methanolic extract of *Ziziphus jujuba* seed powder with bioactive compounds.**

## Feed and water intake of experimental animals

The results indicate that $CCl_4$-induced liver fibrosis negatively impacted feed and water intake in rats, with the Disease control group ($G_1$) consistently having the lowest consumption throughout the experimental period. Treatment with *Z. jujuba* seed powder ($G_2$) and seed oil ($G_3$) moderately improved intake, whereas the combined treatment group ($G_4$) exhibited the greatest restoration of both feed and water consumption, almost reaching or exceeding the levels of the normal control group ($G_0$). These findings suggest that the treatments enhance physiological recovery and alleviate disease-induced anorexia and dehydration, promoting a superior nutritional status and possibly accelerating the healing processes (Fig 4 and Fig 5).

## Effect of *Z. jujuba* powder and oil on Liver function biomarkers

As indicated in Table 5, G1 rats exhibited a marked increase in alanine aminotransferase (ALT), aspartate aminotransferase (AST), alkaline phosphatase (ALP), and total bilirubin levels, confirming hepatocellular damage and cholestasis. The administration of *Z. jujuba* seed powder ($G_2$), seed oil ($G_3$), and their combination ($G_4$) ameliorated these biochemical

**Table 3. Compounds identified in the methanolic extract of *Ziziphus jujuba* seed powder by gas chromatography.**

| Peak # | R. T (min) | Name | Area % |
|---|---|---|---|
| 1 | 8.706 | 2-Decenal<br>2-Cyclohexane-1-ol | 2.29 |
| 2 | 9.475 | 2,4-Decadienal (E,E), 2,4-Decadienal (E,Z) | 0.95 |
| 3 | 10.48 | 2,4-Decadienal, 2,4-Decadienal (E,E) | 1.14 |
| 4 | 16.474 | Cetene | 0.97 |
| 5 | 20.888 | E-15-Heptadecenel, 1-Octadecene, 1-Nonadecene<br>Dinocap II, Binapacryl, Dinocap-I | 1.05 |
| 6 | 23.572 | Hexdecanoic acid, methyl ester<br>Demephion, vamidophion | 1.07 |
| 7 | 24.262 | n-hexadecanoic acid | 2.63 |
| 8 | 27.499 | (9E,11E)-Octadecadienoic acid, 10E,12Z-Octadecadienoic acid, 9,12-Octadecadienoic acid (Z,Z)- | 3.62 |
| 9 | 27.635 | Oleic acid, 9-Octadecenoic acid, (E)-<br>Exaltolide[15-Pentadecanolide], Trichlorfon | 10.61 |
| 10 | 28.024 | Octadecanoic acid, cis-Vaccenic acid, 10E, 12Z-Octadecadienoic acid | 2.38 |
| 11 | 31.923 | Hexandioic acid, bis(2-ethylhex) | 25.31 |
| 12 | 34.169 | Bis(2-ethylhexyl) phthalate, Diamyl phthalate, Di-n-butyl phthalate | 17.59 |
| 13 | 37.319 | 1,4-Benzenedicarboxylic acid, Dicyclohexyl phthalate, Di-n-octyl phthalate | 30.46 |

**Table 4. Compounds identified in the hexane extract of *Z. jujuba* seed oil using gas chromatography.**

| Peak # | R. T (min) | Name | Area % |
|---|---|---|---|
| 1 | 25.581 | Hexadecanoic acid, methyl ester, Pentadecanoic acid, 14-methyl-<br>Vamidothion, Demephion | 4.92 |
| 2 | 24.252 | n-Hexadecanoic acid | 4.23 |
| 3 | 26.780 | 9,12-Octadecadienoic acid (Z,Z)-<br>Methyl 10-trans,12-cis-octadecadienoic acid, 10, 13-Octadecadienoic acid, methyl ester | 4.59 |
| 4 | 26.916 | 9-Octadecenoic acid (Z)-, methyl ester, 7-Octadecenoic acid, methyl ester, 9-Octadecenoic acid, methyl esteer<br>Exaltolide [15-Pentadecanolide] | 9.03 |
| 5 | 27.101 | 12,15-Octadecatrienoic acid, methyl, 9,12,15-Octadecatrienoic acid, m | 0.81 |
| 6 | 27.159 | 9,12,15-Octadecatrienoic acid, methyl | 0.83 |
| 7 | 27.402 | Heptadecanoic acid, 16-methyl-, 2 Methyl stearate, Methyl stearate<br>Demephion | 1.34 |
| 8 | 27.499 | 10E,12Z-Octadecadienoic acid, 2 (9E,11E)-Octadecadienoic acid, 9,12-Octadecadienoic acid (Z,Z)- | 4.85 |
| 9 | 27.626 | Oleic Acid, 9-Octadecenoic acid, (E)-, 6-Octadecenoic acid, (Z)-<br>Exaltolide [15-Pentadecanolide] | 16.06 |
| 10 | 28.024 | cis-Vaccenic acid, trans-13-Octadecenoic acid, cis-13-Octadecenoic acid Exaltolide [15-Pentadecanolide] | 4.30 |
| 11 | 31.913 | Hexanedioic acid, bis(2-ethylhexyl | 20.78 |
| 12 | 34.159 | Phalicacid, Phthalic acid, di(2-propylpentyl phthalate, Bis(2-ethylhexyl) phthalate, Bis(2-ethylhexyl) phthalate<br>Bis(2-ethylhexyl)phthalate, Diisobutyl phthalate, Di-n-butylphthalate | 7.81 |
| 13 | 37.299 | 1,4-Benzenedicarboxylic acid, bi, Bis (2-ethylhexyl) phthalate, Di-n-propyl phthalate, Dicyclohexyl phthalate | 20.43 |

indicators, with the combined treatment showing the most effective reduction in ALT (75.56 U/L), ALP (125.43 U/L), AST (125.76 U/L), and bilirubin (0.40 mg/dL) levels. These results illustrate the potent hepatoprotective effects of *Z. jujuba* formulations in mitigating liver dysfunction and cholestasis caused by fibrosis (Table 5; Fig 6).

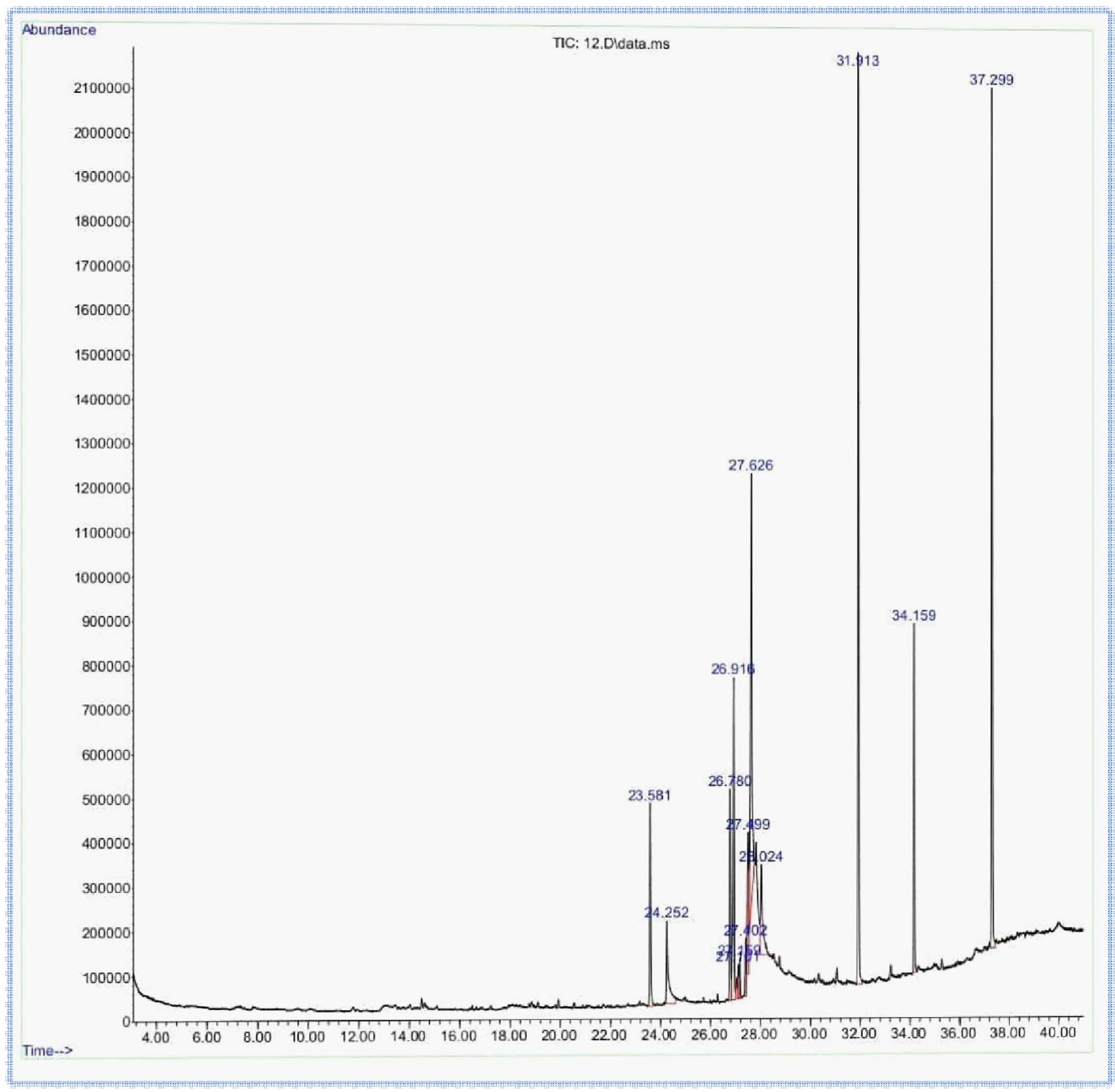

**Fig 3. GC-MS chromatogram of the hexane extract of *Z. jujuba* seed oil with bioactive compounds.**

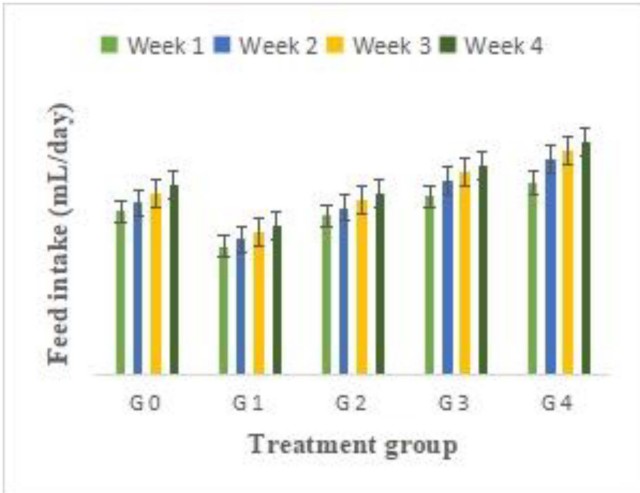

**Fig 4. Effect of *Z. jujuba* seed powder and seed oil supplementation on feed intake (g/day) in the different treatment groups.**

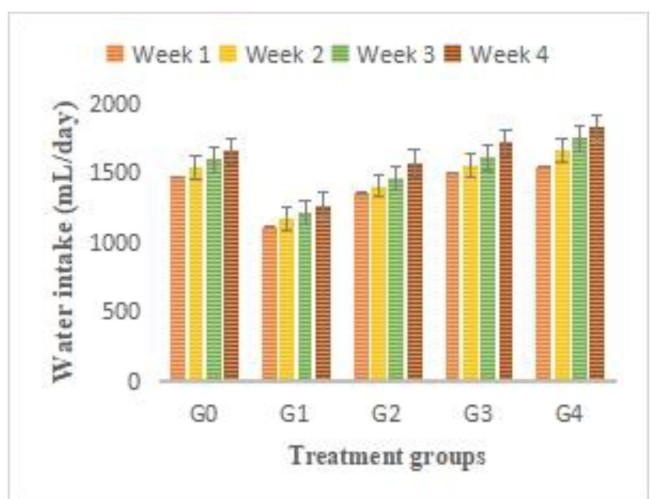

**Fig 5. Effect of *Z. jujuba* seed powder and seed oil supplementation on water intake (mL/day) in the different treatment groups.**

**Table 5. Effect of *Z. jujuba* seed powder and seed oil on liver function biomarkers.**

| Groups | Bilirubin (mg/dL) | ALT (U/L) | AST (U/L) | ALP (U/L) |
|---|---|---|---|---|
| $G_0$ | 0.30±0.01[d] | 35.00±2.92[d] | 80.23±4.15[d] | 131.00±4.93[d] |
| $G_1$ | 3.50±0.14[a] | 165.80±3.82[a] | 250.01±16.23[a] | 200.05±5.76[a] |
| $G_2$ | 1.80±0.13[b] | 95.01±4.08[b] | 160.01±15.16[b] | 173.02±5.51[b] |
| $G_3$ | 0.90±0.01[c] | 90.20±4.42[b] | 140.03±7.35[bc] | 145.76±2.96[c] |
| $G_4$ | 0.40±0.01[d] | 75.56±4.63[c] | 125.76±7.00[c] | 125.43±5.59[d] |
| F. ratio | 674.04*** | 367.61*** | 69.25*** | 688.75*** |

$G_0$, Normal control (no disease induced); which received only the standard basal diet. $G_1$, Disease control group ($CCl_4$-induced), which was diseased and received the standard basal diet; $G_2$ represents the group treated with *Z. jujuba* seed powder (5 g/kg); $G_3$ received Z. jujuba seed oil (5 mL/kg); $G_4$ was given a combination of *Z. jujuba* seed powder (5 g/kg) and seed oil (5 mL/kg) along with the standard, Means having same alphabets in columns do not differ significantly (P-value < 0.05), *** indicates significantly different.

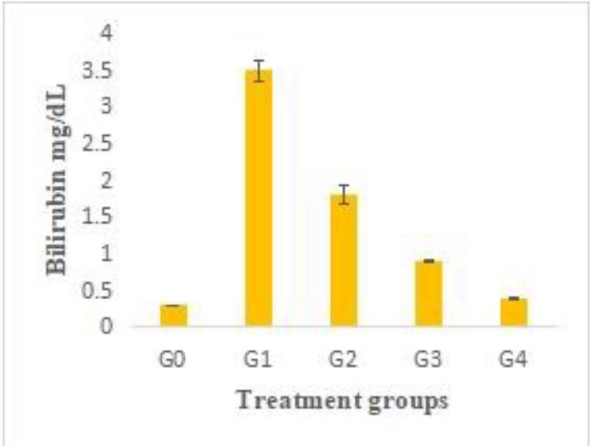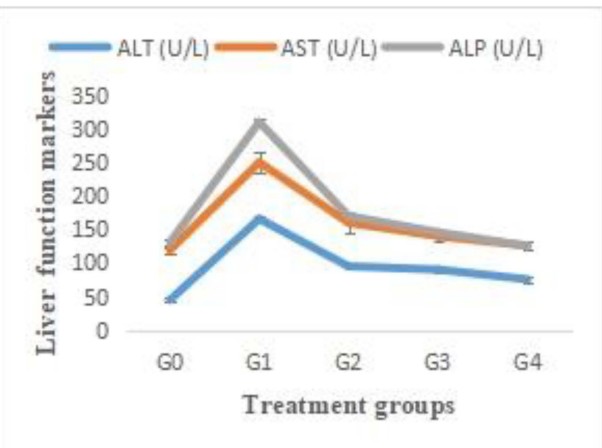

**Fig 6. Effect of *Z. jujuba* seed powder and seed oil on liver function biomarkers: (a) bilirubin, (b) ALT, AST, and ALP in different treatment groups.**

## Oxidative stress and fibrosis biomarkers

CCl$_4$ administration significantly increased malondialdehyde (MDA) levels to 9.15 nmol/g protein in the disease control group (G$_1$) compared to 7.31 nmol/g protein in the normal control (G$_0$), indicating heightened lipid peroxidation and oxidative damage. Nitric oxide (NO) levels, also indicative of oxidative stress, were elevated to 61.34 nmol/g protein in G$_1$ from 28.89 nmol/g protein in G$_0$, while SOD levels were decreased in G$_1$ 56.51 U/mg from 78.37 U/mg in G$_0$. Treatment with *Z. jujuba* seed powder (G$_2$), seed oil (G$_3$), or a combination of both (G$_4$) progressively reduced these markers. G$_4$ showed the greatest reduction in MDA (5.47 nmol/g protein) and NOx (46.81 nmol/g protein) levels and an elevation in SOD (71.69 U/mg), demonstrating effective antioxidant defense. Catalase (CAT) enzyme activity, which was suppressed in G$_1$ (5.51 U/mg protein) relative to G$_0$ (8.37 U/mg protein), was restored by treatments, with G$_4$ achieving 8.69 U/mg protein, was restored by treatments, with G$_4$ achieving 2.67 U/mg protein, suggesting improved enzymatic scavenging of reactive oxygen species, as shown in Fig 7 and Table 6.

## Effect of *Z. jujuba* seed powder and oil on lipid profile and glucose levels

As shown in Table 7, exposure to CCl$^-$ resulted in an increase in serum triglyceride (TG), very low-density lipoprotein (VLDL), and low-density lipoprotein (LDL) levels, whereas high-density lipoprotein (HDL) levels were reduced. Treatment with *Z. jujuba* seed powder (G$_2$) and seed oil (G$_3$) partially reversed these effects, with decreased TG (109.92 and 95.23 mg/dL, respectively) and increased HDL (25.54 and 32.00 mg/dL, respectively). The combined treatment (G$_4$) yielded the most significant improvements, normalizing TG and HDL levels to 80.01 mg/dL and 34.60 mg/dL, respectively, which were closer to the control values. LDL levels were also substantially lower in G$_4$ (22.30 mg/dL) than in G$_1$ (41.09 mg/dL).

Fasting glucose showed minor, statistically nonsignificant variations between groups, with G$_1$ slightly higher (114.23 mg/dL) than G$_0$ (100.00 mg/dL), and G$_4$ nearly restored to normal (98.92 mg/dL). These data suggest that *Z. jujuba* treatment effectively ameliorates lipid abnormalities induced by fibrotic liver damage with a limited impact on glucose homeostasis (Table 7).

## Effect of *Z. jujuba* seed powder and oil on inflammatory markers

CCL$_4$ administration significantly increased the levels of TNF-α to 17.90 pg/mL in the disease control group (G$_1$) compared to 8.03 pg/mL in the normal control (G$_0$), indicating an active immune response, stellate cell activation, and progression toward scar tissue accumulation. IL-6 levels were also slightly elevated to 28.38 pg/mL in G$_1$, from 21.53 pg/mL. Treatment

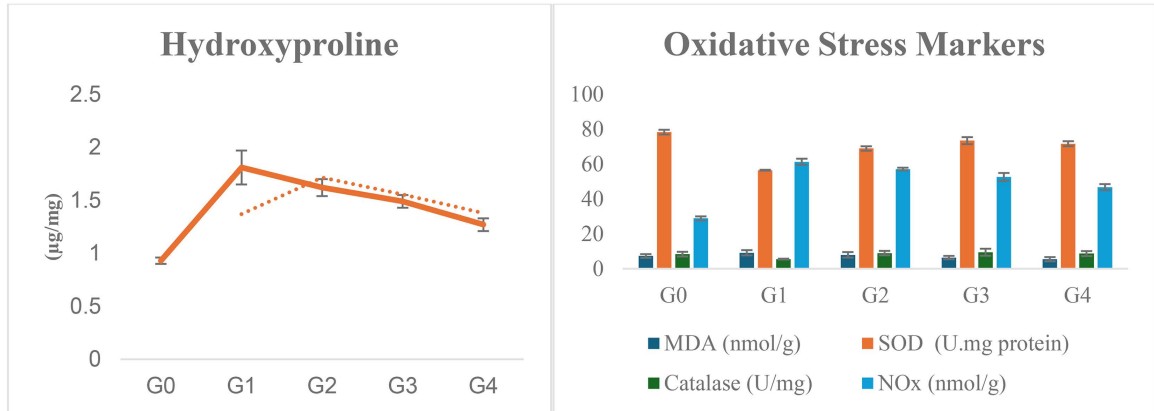

**Fig 7. Effect of *Z. jujuba* seed powder and seed oil on oxidative stress biomarkers: (a) hydroxyproline, (b) MDA, SOD, NOx, Catalase in different treatment groups.**

**Table 6. Effect of *Z. jujuba* seed powder and seed oil on liver oxidative stress and fibrosis biomarkers.**

| Groups | Hydroxyproline (µg/mg) | MDA (nmol/g) | NOx (nmol/g) | SOD (U/mg Protein) | Catalase (U/mg) |
|---|---|---|---|---|---|
| $G_0$ | 0.93 ± 0.03[d] | 7.31 ± 1.08[c] | 28.89 ± 1.12 | 78.37 ± 1.35[a] | 8.37 ± 1.35[b] |
| $G_1$ | 1.81 ± 0.16[a] | 9.15 ± 1.52[a] | 61.34 ± 1.78 | 56.51 ± 0.24[d] | 5.51 ± 0.24[c] |
| $G_2$ | 1.62 ± 0.08[ab] | 7.96 ± 1.59[b] | 57.19 ± 0.75 | 68.97 ± 1.26[c] | 8.97 ± 1.26[ab] |
| $G_3$ | 1.49 ± 0.06[bc] | 6.39 ± 0.97[c] | 52.65 ± 2.28 | 73.46 ± 2.01[ab] | 9.46 ± 2.01[a] |
| $G_4$ | 1.27 ± 0.06[c] | 5.47 ± 1.21[d] | 46.81 ± 1.73 | 71.69 ± 1.43[b] | 8.69 ± 1.43[ab] |
| F. ratio | 42.84*** | 12.48*** | 182.92*** | 142.51** | 25.31*** |

$G_0$, Normal control (no disease induced); which received only the standard basal diet. $G_1$, Disease control group ($CCl_4$-induced), which was diseased and received the standard basal diet; $G_2$ represents the group treated with *Z. jujuba* seed powder (5 g/kg); $G_3$ received Z. jujuba seed oil (5 mL/kg); $G_4$ was given a combination of *Z. jujuba* seed powder (5 g/kg) and seed oil (5 mL/kg) along with the standard basal diet; Hydroxyproline, Collagen biomarker; MDA, Malondialdehyde; NOx, Nitric oxide metabolites; Catalase, Catalase enzyme activity; NS: Non-significant, Means having same alphabets in columns do not differ significantly (P-value < 0.05), *** means significantly different.

**Table 7. Effect of *Z. jujuba* seed powder and seed oil on lipid profiles and glucose levels.**

| Groups | Glucose | TG | TC | HDL | VLDL | LDL |
|---|---|---|---|---|---|---|
| G0 | 100.00 ± 10.68 | 75.00 ± 3.32[d] | 78.27 ± 7.52[b] | 37.00 ± 6.80[a] | 15.00 ± 0.49[d] | 20.00 ± 0.81[d] |
| G1 | 114.23 ± 2.01 | 150.23 ± 15.49[a] | 97.37 ± 1.60[a] | 24.22 ± 1.48[d] | 32.04 ± 0.78[a] | 41.09 ± 2.11[a] |
| G2 | 108.20 ± 3.55 | 109.92 ± 5.48[b] | 72.42 ± 1.90[c] | 25.54 ± 1.28[c] | 21.98 ± 0.89[b] | 24.90 ± 1.86[b] |
| G3 | 102.45 ± 6.84 | 95.23 ± 5.60[bc] | 75.56 ± 3.44[bc] | 32.00 ± 1.04[c] | 19.04 ± 0.62[c] | 24.49 ± 0.87[b] |
| G4 | 98.92 ± 8.76 | 80.01 ± 4.31[c] | 71.90 ± 2.66[d] | 34.60 ± 0.72[b] | 16.42 ± 0.63[c] | 22.30 ± 0.56[c] |
| F. ratio | 2.41[NS] | 41.32*** | 31.05*** | 88.17*** | 226.55*** | 176.74*** |

$G_0$, Normal control (no disease induced); which received only the standard basal diet. $G_1$, Disease control group ($CCl_4$-induced), which was diseased and received the standard basal diet; $G_2$ represents the group treated with *Z. jujuba* seed powder (5 g/kg); $G_3$ received Z. jujuba seed oil (5 mL/kg); $G_4$ was given a combination of *Z. jujuba* seed powder (5 g/kg) and seed oil (5 mL/kg) along with the standard basal diet; TG, Triglycerides; TC, Total cholesterol; HDL, High-density lipoprotein; VLDL, Very low-density lipoprotein; LDL, Low-density lipoprotein; NS: Non-significant, Means having same alphabets in columns do not differ significantly (P-value < 0.05), *** means significantly different.

with *Z. jujuba* seed powder ($G_2$), seed oil ($G_3$), or a combination of both ($G_4$) progressively reduced these markers. $G_2$ showed the greatest reduction in TNF-α (11.21 pg/mL), followed by $G_4$ (12.49 pg/mL). IL-6 levels were also restored by the treatments, with $G_4$ achieving 18.81 pg/mL, suggesting reduced inflammatory signaling and stellate cell activation, as shown in Table 8.

### Effect of *Z. jujuba* seed powder and oil on Serum protein profile

Serum total protein concentration decreased in $G_1$ (5.87 g/dL) versus $G_0$ (6.80 g/dL), reflecting liver synthetic dysfunction. Supplementation with *Z. jujuba* improved protein levels, with $G_4$ reaching 5.98 g/dL, indicating the partial restoration of hepatic biosynthesis. Globulin levels remained relatively unchanged across the groups (Fig 8).

### Effect of *Z. jujuba* seed powder and oil on hematological profile

The hematological data showed a significant reduction in the RBC count in the disease control group ($G_1$:$6.19 \pm 0.33 \times 10^6$/μL) compared to the normal control group ($G_0$:$7.80 \pm 0.35 \times 10^6$/μL), indicating anemia induced by $CCl_4$ toxicity. Hemoglobin

**Table 8. Effect of *Z. jujuba* seed powder and seed oil on inflammation markers.**

| Groups | TNF-α (pg/mL) | IL-6 (pg/mL) |
|---|---|---|
| $G_0$ | 8.03 ± 1.03[d] | 21.53 ± 2.41[c] |
| $G_1$ | 17.90 ± 5.92[a] | 28.38 ± 4.09[a] |
| $G_2$ | 11.21 ± 0.85[c] | 24.72 ± 1.04[b] |
| $G_3$ | 13.98 ± 1.36[b] | 21.47 ± 2.65[c] |
| $G_4$ | 12.49 ± 1.61[bc] | 18.81 ± 2.72[d] |
| F. ratio | 116.53** | 78.41** |

$G_0$, Normal control (no disease induced); which received only the standard basal diet. $G_1$, Disease control group ($CCl_4$-induced), which was diseased and received the standard basal diet; $G_2$ represents the group treated with *Z. jujuba* seed powder (5 g/kg); $G_3$ received Z. jujuba seed oil (5 mL/kg); $G_4$ was given a combination of *Z. jujuba* seed powder (5 g/kg) and seed oil (5 mL/kg) along with the standard basal diet; TNF-α, Tumor necrosis factor-alpha; IL-6, Interleukin-6; NS: Non-significant, Means having same alphabets in columns do not differ significantly (P-value < 0.05), *** means significantly different.

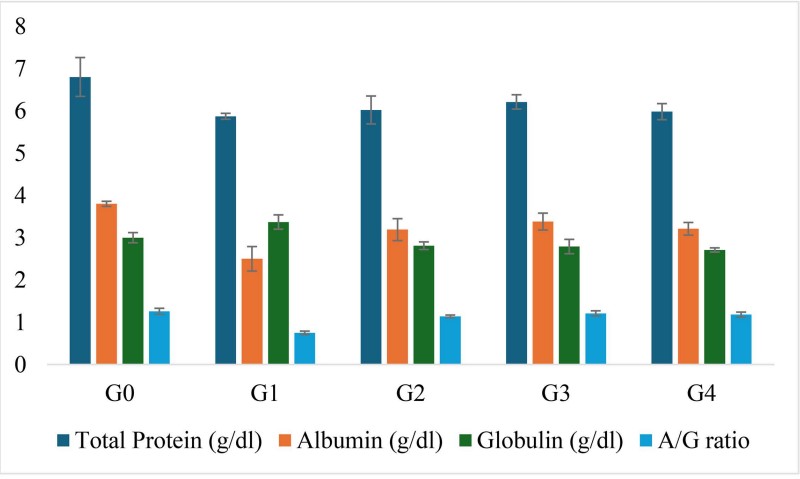

**Fig 8. Effect of *Ziziphus jujuba* seed powder and seed oil on serum protein profile (total protein, Albumin, Globulin, A/G ratio) in different treatment groups.**

levels followed a similar decline in $G_1$ (10.23 ± 0.49 g/dL) compared to $G_0$ (12.50 ± 0.66 g/dL). The hematocrit percentage also significantly decreased in $G_1$ (33.33 ± 1.32%) compared to $G_0$ (43.50 ± 1.93%). Treatment with seed powder ($G_2$), seed oil ($G_3$), and their combination ($G_4$) improved these indices, with $G_4$ nearing normal values (RBC: 7.39 ± 0.59 × 10⁶/µL; Hb: 13.81 ± 0.77 g/dL; HCT: 41.50 ± 1.21%). The mean corpuscular volume, hemoglobin, and hemoglobin concentrations were similar across the groups. These findings reflect the protective effects of *Z. jujuba* interventions on erythropoiesis and overall hematological health in rats with fibrosis (Fig 9).

**Effect of *Z. jujuba* seed powder and oil on differential White blood cell (WBC) count**

Liver fibrosis induced by $CCl_4$ resulted in a significant increase in the total WBC count to 14.22 × 10³/µL in the disease group ($G_1$) compared to 8.32 × 10³/µL in the normal control group ($G_0$). This finding indicates an activated inflammatory response in the cells. Lymphocytes decreased to 45.23% in $G_1$, whereas neutrophils increased to 47.27%, indicating a shift toward neutrophil inflammation. The monocyte count increased to 6.50%. The treatment groups showed improvement with combined treatment ($G_4$); nearly normalizing values WBC count was 8.80 × 10³/µL, lymphocytes 67.98%, neutrophils 26.72%, and monocytes 4.20%. Eosinophil and basophil counts remained unchanged and within the normal range. These changes demonstrate that *Z. jujuba* treatment effectively mitigates the fibrotic inflammatory profile (Fig 10).

## Discussion

This study explored the potential therapeutic effects of *Z. jujuba* seed powder and oil on liver fibrosis in rats induced by carbon tetrachloride ($CCl_4$). The adverse effects of $CCl_4$ on feed and water intake, hematological parameters, liver function, oxidative stress, and fibrosis markers were significantly ameliorated by the treatments, highlighting the hepatoprotective and antifibrotic capabilities of *Z. jujuba*. This discussion integrates these findings with the current literature and elaborates on the mechanistic insights.

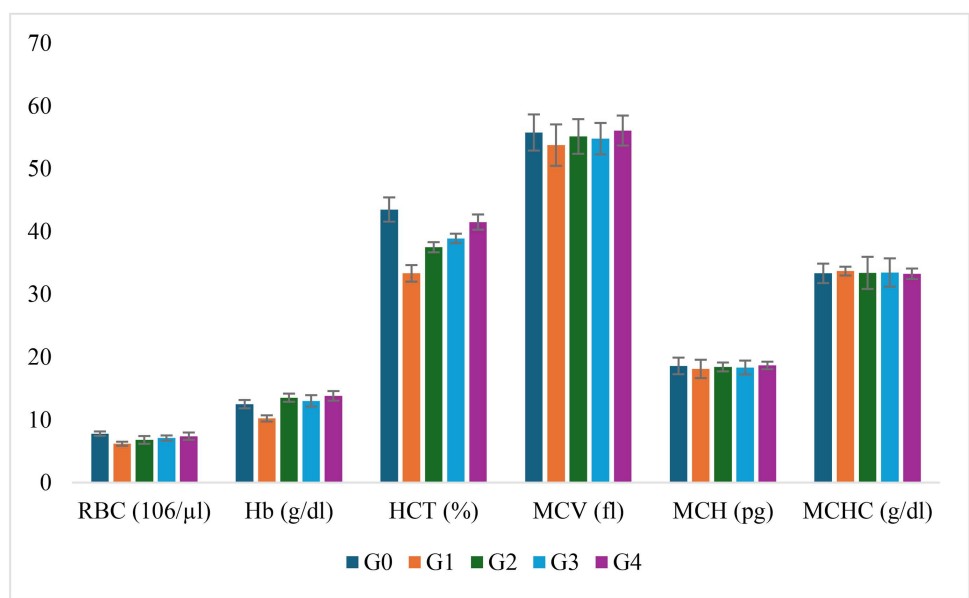

**Fig 9. Effects of *Ziziphus jujuba* seed powder and seed oil on hematological parameters (RBC, Hb, HCT, MCV, MCH, and MCHC) in different treatment groups.**

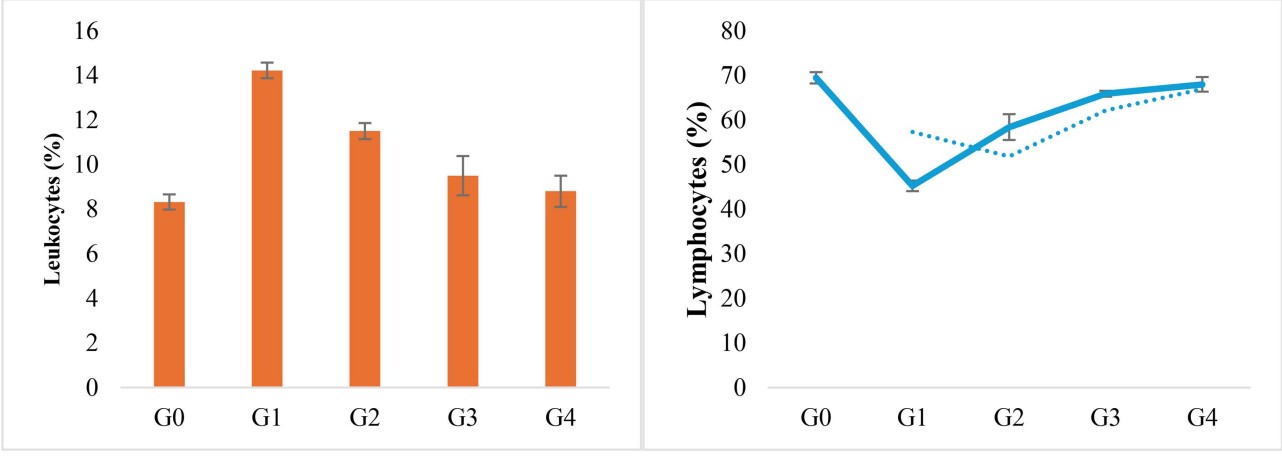

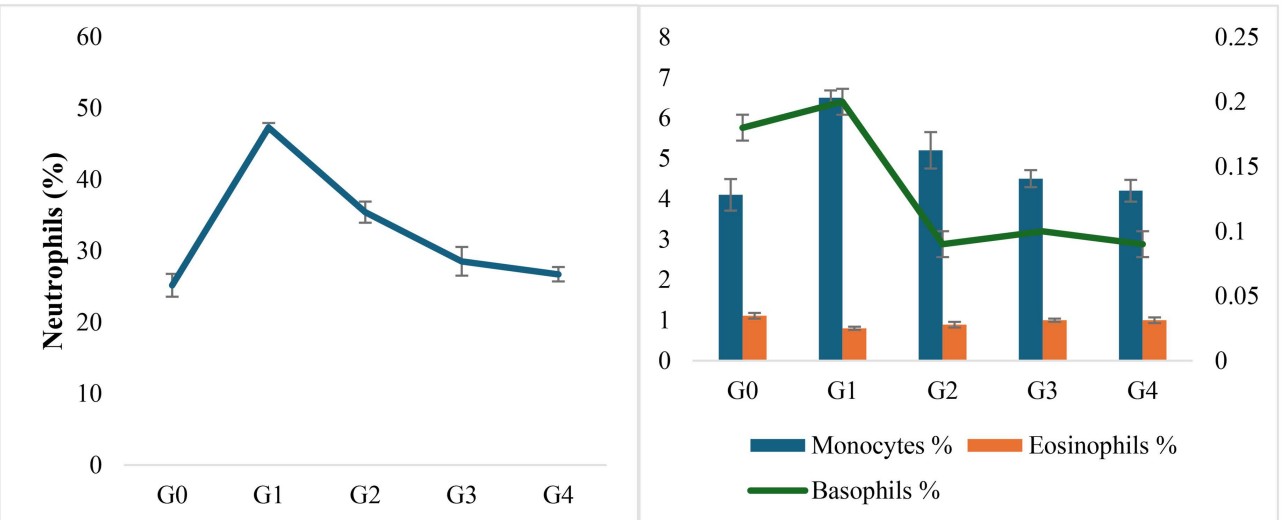

**Fig 10. Effect** *of Z. jujuba* **seed powder and seed oil on (a) leukocyte profile (WBC), (b) lymphocytes, (c) neutrophils, and (d) white blood cell differential counts (monocytes, eosinophils, and basophils) in different treatment groups.**

The acetone and ethanol extracts of the seed powder exhibited the highest antioxidant activity in both DPPH and FRAP assays. Interestingly, Zheng et al. (2022) reported that flavonoid-rich seed extracts attenuate oxidative injury in vivo via $Nrf_2$ activation and suppression of NF-κB signaling [33]. Seed oil ethanolic extracts also demonstrated notable FRAP values despite lower phenolic content, consistent with previous findings, which attributed oil antioxidant effects to tocopherols, phytosterols, and unsaturated fatty acids, similar to our study [34]. Furthermore, a previous study found that lipid-soluble constituents from *Z. jujuba* seeds can enhance antioxidant defenses in vivo, complementing phenolic-driven radical scavenging from the powder [35].

GC-MS analysis of *Z. jujuba* seed powder and seed oil identified several bioactive compounds with potential anti-fibrotic effects relevant to liver fibrosis [36]. Oleic acid, the predominant monounsaturated fatty acid in both extracts, ameliorates liver fibrosis by modulating the NF-κB signaling pathway, which is critically involved in hepatic inflammation and fibrogenesis [37]. NF-κB inhibition leads to reduced activation of HSCs, the key effectors of fibrosis, thereby decreasing

extracellular matrix deposition and inflammatory cytokine production [6,7]. Linoleic acid, another significant unsaturated fatty acid, has been demonstrated to interfere with the transforming growth factor-beta (TGF-β) and Smad signaling cascades, which are central pathways in promoting HSC activation and fibrotic progression in the liver [38]. Through the downregulation of TGF-β1 expression, linoleic acid reduces collagen synthesis and fibrosis advancement. Moreover, n-hexadecanoic acid (palmitic acid) may exert anti-inflammatory effects by repressing profibrotic gene expression, although some studies suggest that it can also promote fibrosis depending on the context, highlighting its complex role [36]. Since, Oleic acid's high presence in the extract may explain the observed reduction in NF-κB signaling, as evidenced by previous studies [36,38].

GC-MS analysis identified bioactive compounds such as oleic acid, linoleic acid, and phthalate derivatives, including bis (2-ethylhexyl) phthalate and hexanedioic acid Bis(2-ethylhexyl) ester. While these compounds are acknowledged for their therapeutic properties, the presence of phthalates raises concerns regarding potential contamination from extraction materials, as phthalates are not naturally found in plants [65]. Phthalates, including Bis (2-ethylhexyl) phthalate, are recognized as environmental contaminants and endocrine disruptors that may interfere with the observed biological effects. These compounds have been shown to affect macrophage activity and fibrogenic signaling, potentially resulting in liver damage. Therefore, it is essential to distinguish the effects of Z. jujuba from the potential impact of phthalate contamination to accurately interpret the results [65–67]. To minimize contamination, future research should employ glassware and solvents that are less susceptible to phthalate introduction and ensure meticulous management of the extraction process. Phthalate derivatives, such as Hexanedioic acid, Bis(2-ethylhexyl) ester in seed oil, have been reported to modulate macrophage activity and fibrogenic signaling in experimental models [39]. These findings warrant a careful evaluation of the therapeutic applications and extraction purity. Overall, the significant active compounds in Z. jujuba seed extracts potentially confer hepatoprotective effects by targeting key fibrotic pathways, such as NF-κB and TGF-β/Smad, which aligns with multiple studies investigating natural product interventions in liver fibrosis [37,38,40]. The promising effects of Z. jujuba seed powder and oil in Sprague Dawley rats highlight the therapeutic potential of the current study, which significantly improved antifibrotic biomarkers, especially hydroxyproline and oxidative stress markers.

The decline in feed and water intake observed in the fibrotic group ($G_1$) reflects systemic toxicity and hepatic dysfunction resulting from the oxidative damage and inflammation. The improvement in intake in the treatment groups, particularly in the combined seed powder and oil group ($G_4$), suggests the mitigation of metabolic stress and restoration of gastrointestinal function. Our observations are consistent with those of Begum et al. (2022), who reported enhanced appetite and hydration in rodents treated with hepatoprotective phytoconstituents [41].

In rats, liver fibrosis caused by $CCl_4$ led to a marked increase in liver enzymes such as ALT, AST, ALP, and bilirubin. However, the administration of Z. jujuba seed powder and oil significantly improved the levels of these enzymes, indicating strong hepatoprotective properties. Comparable findings have been reported in other interventional studies involving Z. jujuba and natural products in liver fibrosis animal models. Li et al. (2023) demonstrated that flavonoids extracted from Z. jujuba protected hepatocytes against acetaminophen-induced liver injury by significantly reducing ALT and AST levels through antioxidant and anti-inflammatory mechanisms [42]. Similarly, Liu et al. (2015) showed that polysaccharides isolated from Z. jujuba markedly suppressed increases in ALT, AST, and ALP in $CCl_4$-injured mice, indicating the mitigation of hepatic damage via the enhancement of antioxidant enzymes and reduction of lipid peroxidation [43]. Furthermore, Z. jujuba seed powder and oil have been shown to significantly downregulate elevated liver enzymes, such as ALT and AST, in vivo, primarily by modulating key signaling pathways, including NF-κB and TGF-β/Smad, which reduce hepatic inflammation and fibrosis progression [44]. This hepatoprotective effect is coupled with antioxidant and anti-inflammatory actions, which collectively improve liver function and structure in fibrosis models. These studies collectively endorse the hepatoprotective and anti-fibrotic potential of Z. jujuba components, consistent with our biochemical findings of decreased serum liver enzyme activity and bilirubin levels in the present study.

*Z. jujuba* seed powder and oil contain significant bioactive compounds, such as oleic acid, linoleic acid, and methyl esters, which play crucial roles in modulating oxidative stress markers, including MDA, NOx, SOD, and CAT, in liver fibrosis. Oleic acid reduces hepatic oxidative stress by activating Nrf2-dependent antioxidant pathways, thereby decreasing MDA levels and restoring SOD and CAT activity [45]. Similarly, linoleic acid supplementation attenuates lipid peroxidation and inflammatory responses by promoting Nrf2 signaling and scavenging reactive oxygen species, consequently lowering MDA and NOx levels and enhancing antioxidant enzyme defense [46]. These actions collectively suppress the oxidative damage implicated in hepatic stellate cell activation and fibrosis progression [47,48]. Thus, the antioxidant properties of *Z. jujuba* constituents likely contribute to the observed normalization of oxidative stress markers in liver fibrosis models, highlighting their therapeutic potential in targeting oxidative stress-linked pathways.

Moreover, the bioactive compounds in *Z. jujuba* seed powder and oil not only play role in modulate oxidative stress markers but also inflammatory markers (TNF-α, IL-6) by reducing hepatic stellate cell activation and ECM deposition. Elevated TNF-α and IL-6 levels in liver fibrosis activate the NF-kB and JAK/STAT pathways, promoting the trans-differentiation of HSCs into myofibroblasts that secrete ECM via TGF-β and JNK signaling [70]. Thus, the anti-inflammatory and hepatoprotective properties of *Z. jujuba* help restore the inflammatory markers to their normal values in a liver fibrosis model [69].

Furthermore, *Z. jujuba* seed powder and oil had a positive impact on the lipid profile by reducing total cholesterol, LDL, and triglyceride levels while improving HDL. These changes likely stem from bioactive compounds such as oleic acid and linoleic acid, which regulate lipid metabolism by modulating the PPARα and AMP-activated protein kinase (AMPK) pathways, thus improving lipid clearance and reducing hepatic steatosis [49]. This lipid-lowering effect has been observed in multiple intervention studies, where natural products rich in unsaturated fatty acids significantly alleviated dyslipidemia associated with liver fibrosis and metabolic dysfunction [36,50].

Regarding hematological parameters, treatment with *Z. jujuba* formulations improved RBC and WBC counts, which are typically suppressed under fibrotic conditions owing to systemic inflammation and oxidative stress. These effects can be attributed to the antioxidant and immunomodulatory compounds in the seed extracts, which enhance hematopoiesis and resolve inflammation by regulating cytokine signaling and oxidative stress pathways [48,51]. Similar improvements in blood cell indices have been documented in preclinical models treated with flavonoid-rich botanicals, reinforcing the role of *Z. jujuba* in restoring hematological homeostasis during liver injury [52–54].

Mechanistically, *Z. jujuba* seed constituents likely interfere with pro-fibrogenic signaling pathways, including TGF-β and hepatic stellate cell activation pathways. The synergism between the water-soluble polyphenols in the seed powder and the lipid-phase antioxidants in the seed oil may account for the enhanced therapeutic efficacy observed with the combination treatment. This integrative phytochemical approach offers comprehensive modulation of fibrosis pathophysiology beyond what single extracts achieve, consistent with the multi-target potential of medicinal plants [55–57].

The primary limitation of this study is the absence of a liver histopathological analysis of the *Z. jujuba* seed powder and seed oil treatment groups, in comparison with a normal control and a disease control group, to accurately assess liver architecture and hepatocyte condition. Additionally, this study did not explore molecular pathways, such as TGF-β and NF-κB protein expression, or clinical validation. The findings are preliminary and based only on a CCl₄-induced animal model. However, these preclinical findings provide compelling evidence supporting *Z. jujuba* seed powder and oil as promising candidates for the adjunctive management of liver fibrosis, especially in high-burden, resource-limited settings, such as Pakistan. Moreover, emerging nanotechnologies, such as nano-emulsions, nanoparticle-based delivery, and nano-encapsulation, represent promising strategies to enhance the bioavailability and targeted delivery of *Z. jujuba* bioactive compounds to hepatic cells. Nano-encapsulation holds great potential for improving specific liver cell targeting, maximizing therapeutic efficacy, and minimizing systemic side effects. Future research should focus on developing and optimizing nanocarrier systems for *Z. jujuba* formulations to realize their full clinical potential in treating liver fibrosis.

## Conclusion

This study revealed that *Z. jujuba* seed powder and oil exhibit hepatoprotective and antifibrotic properties in a rat model of $CCl_4$-induced liver fibrosis. Gas chromatography-mass spectrometry (GC-MS) analysis identified bioactive compounds such as oleic acid, linoleic acid, and various fatty acid methyl esters, which likely contribute to the observed hepatoprotective effects. The combination of seed powder and oil significantly normalized the liver function enzymes (ALT, AST, and ALP) and bilirubin levels, effectively reversing hepatocellular injury and cholestasis. Moreover, *Z. jujuba* treatment improved oxidative stress markers (MDA, NOx, SOD, and CAT activity), suggesting an antioxidant defense mechanism through the modulation of pathways involved in oxidative damage and fibrogenesis. Restoration of the lipid profiles and hematological parameters (RBC and WBC) further underscores the therapeutic potential of the extract in mitigating liver damage and fibrosis. Additionally, *Z. jujuba* reduced TNF-α and IL-6 levels, indicating decreased inflammatory signaling and activation of stellate cells. Future research should explore the molecular mechanisms underlying these effects and confirm their safety and efficacy in larger animal models or clinical settings to validate *Z. jujuba* as a standardized treatment for liver fibrosis and related conditions.

## Supporting information

**S1 File. Details animals based experiments was reporting as per Animal Research Reporting of In vivo Experiments (ARRIVE) guidelines.**
(DOCX)

## Author contributions

**Conceptualization:** Khurram Afzal, Asad Abbas, Adnan Amjad.

**Data curation:** Asad Abbas, Talha Bin Iqbal, Bipindra Pandey.

**Formal analysis:** Talha Bin Iqbal, Naeem Sarwar.

**Funding acquisition:** Khurram Afzal, Asad Abbas, Bipindra Pandey.

**Investigation:** Laiba Tanvir, Naeem Sarwar, Rameen Naeem, Ayman Furqan.

**Methodology:** Laiba Tanvir, Talha Bin Iqbal, Rameen Naeem, Ayman Furqan.

**Project administration:** Khurram Afzal, Asad Abbas, Bipindra Pandey.

**Resources:** Khurram Afzal, Asad Abbas, Bipindra Pandey.

**Software:** Asad Abbas, Adnan Amjad.

**Supervision:** Khurram Afzal, Asad Abbas.

**Validation:** Khurram Afzal, Adnan Amjad, Bipindra Pandey.

**Visualization:** Khurram Afzal, Asad Abbas, Adnan Amjad, Bipindra Pandey.

**Writing – original draft:** Laiba Tanvir, Rameen Naeem, Ayman Furqan.

**Writing – review & editing:** Asad Abbas, Bipindra Pandey.

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
