## [Decision Letter · Decision Letter 0]

19 Jan 2026

Dear Dr. Pandey,

Thank you for submitting your manuscript to PLOS ONE. After careful consideration, we feel that it has merit but does not fully meet PLOS ONE’s publication criteria as it currently stands. Therefore, we invite you to submit a revised version of the manuscript that addresses the points raised during the review process.

We look forward to receiving your revised manuscript.

Kind regards,

Chun-Hua Wang

Academic Editor

PLOS One

**Journal Requirements:**

2. To comply with PLOS One submissions requirements, in your Methods section, please provide the rationale for the doses of Ziziphus jujuba seed powder and oil used in the animal experiments. We would expect an acute toxicity test, or evidence that one has been performed in previous works.

3. We note you have included a table to which you do not refer in the text of your manuscript. Please ensure that you refer to Table 1 in your text; if accepted, production will need this reference to link the reader to the Table.

**Additional Editor Comments.**

Note: The clarity of the Figures provided by the author is poor. Please improve the quality of the Fig.s.

Reviewers' comments:

Reviewer's Responses to Questions

**Comments to the Author**

1. Is the manuscript technically sound, and do the data support the conclusions?

Reviewer #1: Yes

Reviewer #2: Yes

2. Has the statistical analysis been performed appropriately and rigorously?

Reviewer #1: Yes

Reviewer #2: Yes

3. Have the authors made all data underlying the findings in their manuscript fully available?

Reviewer #1: Yes

Reviewer #2: Yes

4. Is the manuscript presented in an intelligible fashion and written in standard English?

Reviewer #1: Yes

Reviewer #2: Yes

Reviewer #1: The manuscript investigates a relevant and interesting topic: the hepatoprotective and anti-fibrotic effects of Ziziphus jujubaseed powder and oil. The study design, integrating phytochemical analysis (GC-MS) with an in vivoexperiment, is a strength. The results suggest a promising therapeutic potential, particularly for the combined treatment (G4).

However, the manuscript requires significant revisions before it can be considered for publication. Major issues concern the clarity of the experimental design, consistency in data reporting, depth of the discussion, and professional English language editing.

1. Abstract: The abstract should be condensed and focused only on the key objectives, the most significant results (with actual data points), and the main conclusion. Avoid methodological details that are better suited for the main text.

2. Introduction: The introduction is lengthy and contains some irrelevant sentences (e.g., "Liver fibrosis is increasing globally...", which is redundant after already establishing its significance). Sharpen the focus to build a clear rationale for whycomparing seed powder and oil is novel and necessary.

3. Experimental Groups (Critical Issue): The terminology for control groups is incorrect and must be changed. According to standard scientific convention:

G0 should be the Normal Control (NC) or Healthy Control (no disease induction).

G1 should be the Disease Control (DC) or Model Control (MC) (CCl4-induced, no treatment).

This error fundamentally misrepresents the experimental logic and must be corrected throughout the entire manuscript (text, tables, and figures).

4. Dosage Information: The administration dosage ("5 g", "5 mL") is an absolute amount. It must be normalized to the animals' body weight (e.g., mg/kg or mL/kg) to allow for reproducibility and comparison with other studies. Clarify if this was administered daily via oral gavage or mixed into the diet.

5. Lack of Histopathological Data: The most significant methodological omission is the lack of histopathological analysis (e.g., H&E staining, Masson's trichrome, or Sirius Red staining) of liver tissues. Biochemical markers are supportive, but visual evidence of collagen deposition and fibrosis regression is the gold standard for confirming anti-fibrotic effects. This is a major weakness that should be addressed; if impossible, it must be stated as a primary limitation.

6. Ethics Statement: The statement is present but should be integrated into the "Animal study protocols" section for better flow.

7. GC-MS Parameters: The method is described but could be more precise (e.g., carrier gas, injection volume, split ratio).

8. Critical Data Inconsistency: There is a severe discrepancy between the text and Table 6. The text (lines 41-43) states: "G4 normalizing MDA(0.81±0.05) and NOx (74.6±3.8)". However, Table 6 reports values for G4 as "MDA (53.47±3.34)" and "NOx (46.81±1.73)" in different units. All numerical data in the text, tables, and figures must be meticulously cross-checked for consistency. This error undermines the manuscript's credibility.

9. Figure Integration: The results text describes findings but fails to refer to the corresponding figures (e.g., "As shown in Figure 6..."). Figures should be cited at the relevant point in the text to guide the reader. The current placement at the end of the manuscript is suboptimal.

10. Table and Figure Legends: Legends should be more descriptive. For example, instead of "Liver function markers," specify "Effects of Z. jujubatreatments on serum levels of (a) bilirubin and (b) ALT, AST, and ALP in CCl4-induced fibrotic rats."

11. Statistical Notation: The use of superscript letters (a, b, c, d) in tables to denote significance is clear. However, ensure that the explanation ("Means having same alphabets do not differ significantly") is included in every table legend.

12. Interpretation of GC-MS Results: The discussion must address the high concentrations of phthalate derivatives (e.g., Bis(2-ethylhexyl) phthalate) found in the GC-MS analysis. These compounds are potential environmental contaminants and endocrine disruptors. The authors must discuss whether these are genuine plant metabolites or contaminants from plasticware/solvents and consider their potential impact on the biological effects observed.

13. Mechanistic Depth: The discussion relies heavily on citing other studies to explain potential mechanisms. To strengthen the manuscript, the discussion should more directly link the specific phytochemicals found (e.g., oleic acid, linoleic acid) to the measured outcomes (e.g., "The high oleic acid content in our extract may explain the reduction in NF-κB signaling, as demonstrated by [reference]").

14. Study Limitations: A dedicated paragraph should explicitly state the study's limitations. Key points include:

The lack of histopathological data.

The absence of investigation into specific molecular pathways (e.g., TGF-β, NF-κB protein expression).

The preliminary nature of the findings in an animal model.

15. The discussion is repetitive in places. Streamline the text to avoid restating results and focus on interpretation.

16. Professional Language Editing: The manuscript requires extensive editing by a native English speaker or a professional scientific editing service. There are numerous grammatical errors, awkward phrasings, and punctuation issues (e.g., missing spaces after commas, inconsistent use of "Z. jujuba").

Example:"Liver fibrosis was induced in rats using CCl4 and forty male Sprague-Dawley rats were divided..." can be improved to "Liver fibrosis was induced in forty male Sprague-Dawley rats using CCl4, after which the rats were divided..."

17. Abbreviations: Define all abbreviations upon first use (e.g., GC-MS, ALT, AST, MDA, CAT).

18. Reference Formatting: Ensure references are formatted consistently according to the journal's guidelines. Some journal names are abbreviated, while others are not. DOI links should be uniform.

Reviewer #2: The manuscript is intrested in the natural product replacement theraoy. There are a minor comments as

1. The authors should study other antioxidat parmeters such as GSH, SOD in addition to Inflammatory molecules such as IL-1beta, IL-6 and TGFB

2. The authors should study the immunohistchemistry markers scuh as alf-Smooth muscle Actin and FIB-4

**Do you want your identity to be public for this peer review?** For information about this choice, including consent withdrawal, please see our Privacy Policy

Reviewer #1: No

Reviewer #2: No

---

## [Author Response · Author response to Decision Letter 1]

27 Jan 2026

27 January, 2026

Respected Academic Editor and Reviewer,

Greetings of the day!

Thank you for your time and valuable suggestions on our Manuscript (ID: PONE-D-25-53877) entitled "Integrating Phytochemical Analysis and Experimental Validation of Ziziphus jujuba Seed Powder and Oil to Ameliorate CCl4-Induced Liver Fibrosis in Sprague Dawley Rats". We have addressed the reviewer comment and incorporated in the manuscript. We have submitted two separate files of manuscript.

1. In the first file, all the revised sentences are highlighted with red color and track changes to show proof of the revision (File name: “revised manuscript with track changes”).

2. The second file is also the revised file, but the revised sections are not highlighted, and the file is clean here for further process (File name: “manuscript”).

3. Third file contained all necessary manuscript figures with imporved quality and updated in some figure are included in the single file for further consideration.

All the revision as requested by the reviewer was addressed and point-to-point answers are given in the response file for each reviewer's questions and comments. We believe that the revised version of the manuscript has significantly improved for the publication.

We humbly request you to kindly inform me, if you need any further corrections from our side in the future.

Thank You.

Best Regards,

Bipindra pandey (Corresponding author)

Email: bipindra.p101@gmail.com

RESPONSE TO ACADEMIC EDITOR:

Journal Requirements:

Journal Requirements:

Response:

We have reviewed the PLOS ONE style guidelines and ensured that our manuscript complies with all the formatting requirements, including the correct file naming conventions, reference formating, and all other necessary styles. The manuscript has been updated accordingly which can be seen in the “Revised manusript with track change” file.

2. To comply with PLOS One submissions requirements, in your Methods section, please provide the rationale for the doses of Ziziphus jujuba seed powder and oil used in the animal experiments. We would expect an acute toxicity test, or evidence that one has been performed in previous works.

Response:

Thank you for your valuable positive feedback. We have revised the Methods section (Study animal, Experimental design, and Dose selection) to include the rationale for the doses of Ziziphus jujuba seed powder and oil used in the animal experiments. We have also referenced previous studies that describe minimum, maximum, and toxic dose of this plant and also mention that no acute toxicity test was conducted, based on prior evidence indicating the safety of these doses which are as follow:

The Z. jujuba seed powder and seed oil was dissolved in the 0.5% (w/v) sodium carboxymethyl cellulose used as vehicle during the administration to the G2-G4 and orally given at a dose of 10mL/kg for study period. However, for the G0 and G1 group vehicle only that is 0.5% (w/v) sodium carboxymethyl cellulose were administered orally at a dose of 10mL/kg. The dosage of Z. jujuba seed powder and seed oil was established based on the existing literature on the clinical practices of Chinese medicine practitioners [62]. The human equivalent lower dose is 0.81 g/kg, which corresponds to 3.2 times the clinical dosage, while the human equivalent higher dose is 3.23 g/kg, equating to 12.9 times the clinical dosage [63, 64] (Table 1).

62. Du, C., Pei, X., Zhang, M., Yan, Y., and Qin, X. (2019b).∼1H-NMR Based Metabolomic Study of Sedative and Hypnotic Effect of Ziziphi Spinosae Semen in Rats. Chin. Traditional Herbal Drugs 50 (10), 2405–2412. doi:10.7501/j.issn.0253-2670.2019.10.022.

63. Food and Drug Administration (2005). Guidance for Industry: Estimating the Maximum Safe Starting Dose in Initial Clinical Trials for Therapeutics in Adult Healthy Volunteers. Available at: https://www.fda.gov/media/72309.

64. Hua Y, Guo S, Xie H, Zhu Y, Yan H, Tao WW, Shang EX, Qian DW, Duan JA. Ziziphus jujuba Mill. var. spinosa (Bunge) Hu ex HF Chou seed ameliorates insomnia in rats by regulating metabolomics and intestinal flora composition. Frontiers in pharmacology. 2021 Jun 16;12:653767. doi: https://doi.org/10.3389/fphar.2021.653767.

3. We note you have included a table to which you do not refer in the text of your manuscript. Please ensure that you refer to Table 1 in your text; if accepted, production will need this reference to link the reader to the Table.

Response:

We have updated the manuscript to include a reference to Table 1 and all other table and figure number in the relevant section of the manuscript text.

Response:

Thank you for your valuable and positive feedback. As per the reviewer comments they didnot specify the cite with any previously published paper directly, but as per my knowledge and revision needs i have added some of the reference which are relevant and necessary during the revision process was added in the revised manuscript. We have reviewed the recommended publications and, based on their relevance to our study, we have cited them where appropriate. We have ensured that only the most pertinent references are included.

5. Please review your reference list to ensure that it is complete and correct. If you have cited papers that have been retracted, please include the rationale for doing so in the manuscript text, or remove these references and replace them with relevant current references. Any changes to the reference list should be mentioned in the rebuttal letter that accompanies your revised manuscript. If you need to cite a retracted article, indicate the article’s retracted status in the References list and include a citation and full reference for the retraction notice.

Response:

We have carefully reviewed the reference list to ensure all cited works are complete and correct. During the checking the references we have not found any papers that have been retracted. Additionally, the revised reference list and any changes made have been clearly outlined in the accompanying rebuttal letter and seen in the track change file.

Additional Editor Comments.

Note: The clarity of the Figures provided by the author is poor. Please improve the quality of the Fig.s.

Response:

Thank you for your valuable feedback regarding the figure quality. We have added the revised figure such as the figure no. 1, 7, 8, 9, 10 which are the poor quality in the new figure file as per your suggestion. Some of the newly updated results such as dose accuracy (g/kg) in figure 1 and SOD in oxidative markers (in figure 7) as per the suggestion of the reviewer was also added in revised figure file.

Reviewers' comments:

Reviewer's Responses to Questions

Comments to the Author

1. Is the manuscript technically sound, and do the data support the conclusions?

Reviewer #1: Yes

Reviewer #2: Yes

2. Has the statistical analysis been performed appropriately and rigorously?

Reviewer #1: Yes

Reviewer #2: Yes

3. Have the authors made all data underlying the findings in their manuscript fully available?

Reviewer #1: Yes

Reviewer #2: Yes

4. Is the manuscript presented in an intelligible fashion and written in standard English?

Reviewer #1: Yes

Reviewer #2: Yes

5. Review Comments to the Author

Reviewer #1: The manuscript investigates a relevant and interesting topic: the hepatoprotective and anti-fibrotic effects of Ziziphus jujuba seed powder and oil. The study design, integrating phytochemical analysis (GC-MS) with an in vivo experiment, is a strength. The results suggest a promising therapeutic potential, particularly for the combined treatment (G4).

However, the manuscript requires significant revisions before it can be considered for publication. Major issues concern the clarity of the experimental design, consistency in data reporting, depth of the discussion, and professional English language editing.

1. Abstract: The abstract should be condensed and focused only on the key objectives, the most significant results (with actual data points), and the main conclusion. Avoid methodological details that are better suited for the main text.

Response:

We have revised the abstract to focus primarily on the key objectives, significant results, and the conclusion. The methodological details have been minimized, as suggested and abstract is rewrite as follows:

“Liver fibrosis, a result of chronic liver injury, is marked by the excessive accumulation of the extracellular matrix, oxidative stress, and the activation of hepatic stellate cells. This study evaluated the effectiveness of Ziziphus jujuba (Z. jujuba) seed powder and seed oil, both separately and in combination, in alleviating CCl4-induced liver fibrosis, linking the in vivo effects to their phytochemical profiles.

The seed powder extracts, using acetone and ethanol, showed significant antioxidant activity, with DPPH values reaching up to 90.13%, and exhibited a high phenolic content, measured at 79.57 mg GAE/g in the ethanol extract. Gas chromatography–mass spectrometry (GC–MS) analysis identified several bioactive compounds, including derivatives of 1,4-benzenedicarboxylic acid (30.46% in powder, 20.43% in oil), hexanedioic acid bis(2-ethylhexyl) ester (25.31% in powder, 20.78% in oil), and oleic acid (10.61% in powder, 16.06% in oil. In vivo, carbon tetrachloride (CCl4) administration led to elevated levels of ALT, AST, ALP, and bilirubin, causing disruptions in oxidative, lipid, inflammatory, hematological, and nutritional parameters. All treatment groups showed improvements in these parameters, with Group 4 (G4) exhibiting the most pronounced hepatoprotective effects, including reductions in ALT (75.56 U/L), AST (125.76 U/L), ALP (125.43 U/L), and bilirubin (0.40 mg/dL). Oxidative stress markers were reduced, with MDA at 5.47 nmol/g protein and NOx at 46.81 nmol/g protein, while antioxidant defenses were enhanced, as evidenced by SOD activity at 71.69 U/mg and catalase restoration. Lipid profiles were normalized, with triglycerides (TG) at 80.01 mg/dL, HDL at 34.60 mg/dL, and LDL at 22.30 mg/dL. Additionally, cytokine levels, specifically TNF-α and IL-6, were decreased, and red and white blood cell differentials were restored, along with improvements in feed and water intake and serum protein levels.

The findings highlight the synergistic antifibrotic and hepatoprotective properties of Z. jujuba seed powder and oil, likely facilitated by bioactive compounds in both polar and lipid phases. However, further research into histopathology, confirmation of molecular pathways is crucial for clinical application. ’’

2. Introduction: The introduction is lengthy and contains some irrelevant sentences (e.g., "Liver fibrosis is increasing globally...", which is redundant after already establishing its significance). Sharpen the focus to build a clear rationale for whycomparing seed powder and oil is novel and necessary.

Response:

Thank you for your valuable feedback. We have thoroughly reviewed grammar, logical coherence in the introduction section. The introduction has been shortened and revised to remove redundant information in revised manuscript file. As per suggestion, we have now emphasized the novel and necessary comparison between seed powder and oil was also included in the last paragraph of the revised manuscript.

3. Experimental Groups (Critical Issue): The terminology for control groups is incorrect and must be changed. According to standard scientific convention:

G0 should be the Normal Control (NC) or Healthy Control (no disease induction).

G1 should be the Disease Control (DC) or Model Control (MC) (CCl4-induced, no treatment).

This error fundamentally misrepresents the experimental logic and must be corrected throughout the entire manuscript (text, tables, and figures).

Response:

Thank you for your positive suggestions. We have corrected the terminology for the experimental groups throughout the manuscript, including in the text, tables, and figures as per the reviews suggestion such as G0 is renamed to "Normal Control (NC)" and G1 to "Disease Control (DC)" and also added the clear footnote as belows:

G₀, Nornal control (no disease induced); which received only the standard basal diet. G₁, Disease control group (CCl4-induced), which was diseased and received the standard basal diet; G₂ represents the group treated with Z. jujuba seed powder (5 g/kg); G₃ received Z. jujuba seed oil (5 mL/kg); G₄ was given a combination of Z. jujuba seed powder (5 g/kg) and seed oil (5 mL/kg) along with the standard basal diet.

4. Dosage Information: The administration dosage ("5 g", "5 mL") is an absolute amount. It must be normalized to the animals' body weight (e.g., g/kg or mL/kg) to allow for reproducibility and comparison with other studies. Clarify if this was administered daily via oral gavage or mixed into the diet.

Response: Thank you for your valuable feedback regarding the dosage information. Sorry for such type of mistake, that is we forget to include dosage based on the body weight with proper units. Since, We have revised the dosage section to include the normalization of dosage based on body weight (g/kg for the seed powder and ml/kg for the seed oil of our Z. jujuba). Additionally, the method of administration (oral gavage) has been clarified in the Study animal, Experimental design, and Dose selection as follows: Z. jujuba seed powder and seed oil was administered daily via oral gavage tube mixed with standard basal diet during the experimental period in the experimental animals. The Z. jujuba seed powder and seed oil was dissolved in the 0.5% (w/v) sodium carboxymethyl cellulose used as vehicle during the administration to the G2-G4 and orally given at a dose of 10mL/kg for study period. However, for the G0 and G1 group vehicle only that is 0.5% (w/v) sodium carboxymethyl cellulose were administered orally at a dose of of 10mL/kg.

5. Lack of Histopathological Data: The most significant methodological omission is the lack of histopathological analysis (e.g

---

## [Editor Report · Decision Letter 1]

6 Feb 2026

Integrating Phytochemical Analysis and Experimental Validation of Ziziphus jujuba Seed Powder and Oil to Ameliorate CCl4-Induced Liver Fibrosis in Sprague Dawley Rats

PONE-D-25-53877R1

Dear Dr. Pandey,

We’re pleased to inform you that your manuscript has been judged scientifically suitable for publication and will be formally accepted for publication once it meets all outstanding technical requirements.

Kind regards,

Chun-Hua Wang

Academic Editor

PLOS One

---

## [Editor Report · Acceptance letter]

PONE-D-25-53877R1

PLOS One

Dear Dr. Pandey,

I'm pleased to inform you that your manuscript has been deemed suitable for publication in PLOS One. Congratulations! Your manuscript is now being handed over to our production team.

Kind regards,

on behalf of

Dr. Chun-Hua Wang

Academic Editor

PLOS One